# Congruence between preferred and actual place of death and its association with quality of death and dying in advanced cancer patients: A nationwide survey in Japan

Mariko Shutoh[1]*, Tatsuya Morita[2], Maho Aoyama[3], Yoshiyuki Kizawa[4], Yasuo Shima[5], Mitsunori Miyashita[3]

1 Minato Home Care Clinic, Chuo-ku, Tokyo, Japan, 2 Department of Palliative and Supportive Care, Seirei Mikatahara General Hospital, Hamamatsu, Shizuoka, Japan, 3 Department of Palliative Nursing, Health Sciences, Tohoku University Graduate School of Medicine, Sendai, Miyagi, Japan, 4 Department of Palliative Medicine, Department of Palliative and Supportive Care, Institute of Medicine, University of Tsukuba, Tsukuba, Ibaraki, Japan, 5 Department of Palliative Medicine, Tsukuba Medical Center Foundation, Tsukuba Medical Center Hospital, Tsukuba, Ibaraki, Japan

* shut44105m@gmail.com

## Abstract

### Background

Satisfying patients' preferences is an important outcome in palliative care. Previous research has reported that a patient's place of death was associated with quality of death and dying.

### Purpose

This study aimed to evaluate the association between the congruence between a patient's preferred and actual place of death and their quality of death and dying, as perceived by their family caregivers.

### Method

Data were obtained from a nationwide cross-sectional questionnaire survey of bereaved family caregivers of patients with cancer in Japan. A total of 13,711 family caregivers participated. We evaluated the quality of death and dying using the Good Death Inventory.

### Results

9,123 responses were analyzed (effective response rate: 67%). Patients who died in their preferred place were categorized as the "achieved group," whereas patients who died in a place that they did not prefer were classified as the "not-achieved group." Good Death Inventory scores were significantly higher for the achieved group

**Data availability statement:** All relevant data are within the manuscript and its Supporting Information files.

**Funding:** The author(s) received no specific funding for this work.

**Competing interests:** The authors have declared that no competing interests exist.

compared with the not-achieved group (48.8 ± 10.1 and 44.0 ± 9.5, respectively; p < 0.001). A multiple linear regression analysis indicated that congruence between the preferred and actual place of death was an independent determinant of good quality of death and dying (p < 0.001).

## Conclusion

Congruence between a patient's preferred and actual place of death may contribute to better quality of death and dying among terminally ill patients with cancer. Congruence between the preferred and actual place of death should be regarded as an essential component in end-of-life care.

## Introduction

The Japanese Ministry of Health, Labor and Welfare has strongly supported the dissemination of specialized palliative care in Japan as part of the National Cancer Program. The ministry also supports institutional palliative care services, which have been covered by National Medical Insurance since 1991. Palliative care units (PCUs) aim to provide symptom control and end-of-life care, mainly for cancer patients and their families. In Japan, approximately 370,000 people die of cancer each year. Of these, only 11.8% die at home, 16.5% die in a PCU, and 68.1% die in hospital [1]. Given that the majority of cancer deaths occur in general hospital wards, palliative care consultation services have been covered by National Medical Insurance since 2002, when designated cancer hospitals were required to establish inpatient palliative care teams.

Previous research indicates that the conceptualization of a good death includes dying in a favorite place [2–3]. Moreover, congruence between the preferred and actual place of death is internationally considered to be a quality indicator of end-of-life care [4]. A previous study that investigated the place of death for people with potential palliative care needs in 14 countries reported that between 12% to 57% died at home, whereas 26% to 87% died in hospital [5]. In the United States, Canada, United Kingdom, Sweden, and the Netherlands, deaths at home are on the rise [6–10]. However, the majority of people in other countries die in hospital wards [5].

More than half of surveyed populations would prefer to be cared for and die at home, and the quality of end-of-life care is deemed better at home than in hospital [11–14]. Additionally, more than half of patients with terminal cancer prefer home hospice care but most die in hospital wards. The reasons are complicated, with many different determinants affecting decisions considering where end-of-life care is provided [15–19].

To our knowledge, the effects of congruence between preferences for and actual place of death on quality of patient death and dying have not yet been elucidated sufficiently in cancer patients. The aim of this study was to evaluate the association between this congruence and cancer patients' quality of death and dying, as perceived by their family caregivers. We hypothesized that cancer patients who died at

their preferred place had better ~~end-of-life~~ quality of death and dying compared with cancer patients who were not able to be in their preferred place when they died.

## Methods

This was a nationwide, cross-sectional, anonymous, self-report questionnaire survey for bereaved family members of deceased cancer patients. This study was one part of the Japan Hospice and Palliative Care Evaluation Study 3 (J-HOPE3). The detailed methodology used in this survey has been described previously [20].

### Participating institutions

We sent letters to 396 institutions that, prior to July 1, 2013, were members of Hospice Palliative Care Japan (HPCJ), an organization of palliative care institutions in Japan. Of these institutions, 49 were acute-care hospitals, 296 were inpatient PCUs, and 51 were home hospice services. We received valid responses from 175 institutions comprising 20 acute hospitals, 133 PCUs, and 22 home hospice services.

### Participants

We conducted a cross-sectional, anonymous, self-report questionnaire between May and July 2014. To identify potential patients, we asked each institution to identify and list up to 80 bereaved family members of patients who had died prior to February 2014. On the basis of previous studies, we considered that 3–12 months after bereavement may be an appropriate time frame for inclusion criteria in terms of both recall bias and the grieving process [20–24]. The inclusion criteria were: (1) the patient died of cancer; (2) the patient was at least 20 years old; and (3) the bereaved family member was at least 20 years old. The exclusion criteria were: (1) the patient received palliative care for less than 3 days; (2) the bereaved family member could not be identified; (3) treatment-associated death or death occurred in an intensive care unit; (4) the potential participant was at risk of suffering serious psychological distress, as determined by the primary physician and a nurse; and (5) the potential participant was incapable of completing the self-report questionnaire because of cognitive impairment or visual disability. The questionnaire was sent to the bereaved family members from each participating institution along with a letter explaining the survey. The return of a completed questionnaire was considered consent to participate in the study. Participants were asked to return their completed questionnaires to the secretariat office (Tohoku University) within 2 weeks. We sent a reminder to non-responders 1 month after sending the questionnaire. If they did not wish to participate in the study, they were asked to check a "no participation" box and return the incomplete questionnaire. Ethical approval for the study was granted by the institutional review boards of Tohoku University and all participating institutions.

### Questionnaires

**Congruence between preferred and actual place of death.** Congruence between preferred and actual place of death was retrospectively assessed by asking family caregivers, "In which place of care did he or she wish to spend his or her last days?" and "In which place of care did you wish [the patient] to spend his or her last days?"

Participants were asked to choose a preferred place for end-of-life care and death from six options: a home setting, an acute-care hospital unit, a palliative care unit, others (e.g., nursing home), no preference, and preference unknown.

We adopted this question on the basis of previous literature and expert opinions [12,25–27].

The questionnaires were sent by the medical institutions that participated in the study and certified each patient's death. By linking the data obtained from medical institutions with the returned questionnaires, we were able to identify the place of death. Patients who died in their preferred place were categorized as the "achieved group." Patients who died in a place that was not their preference were classified as the "not-achieved group."

**Good Death Inventory-short version.** We used the short version of the Good Death Inventory (GDI) to measure patients' achievement of a good death from the perspective of their bereaved family members. This measure was developed on the basis of qualitative interviews and a large-scale quantitative study, and has 18 domains representing concepts important to good death for Japanese patients with cancer. In this study, we used the short version of the GDI, which consists of 10 core items and has sufficient reliability and validity [28,29]. The caregiver was asked to rate each item using 7-point Likert scale (ranging from 1: totally disagree to 7: absolutely agree). Total scores were calculated by summing the scores for all items, with a high total score indicating the achievement of a good death. The GDI has been validated outside Japan [30].

**Covariates.** Bereaved family caregivers were asked to report potential confounding factors that might be associated with the congruence between preferred and actual place of death and patients' quality of death, for example: (1) Presence of other caregivers; (2) Communication with the patient about the disease and about life; and (3) Implementation of end-of-life discussions. We adopted these covariates on the basis of previous literature and expert opinions [12,25–27].

## Analysis

To define the potential determinants of congruence between preferred and actual place of death for patients with cancer, the dependent variables were split into two categories: the achieved group and the not-achieved group. The differences between the groups were determined using Chi-square tests.

T-test and analysis of variance (ANOVA) models were used to examine relationships between the congruence between patients' preferred and actual place of death and their quality of death. Where significant differences were observed, a post hoc Dunnett test was used to explore between which groups the differences lay.

We also conducted linear regression analyses to evaluate the associations between the congruence between patients' preferred and actual place of death and quality of death and dying.

Significance was set at $p < 0.05$. All statistical analyses were performed with EZR (Saitama Medical Center, Jichi Medical University, Saitama, Japan), which is a graphical user interface for R (The R Foundation for Statistical Computing, Vienna, Austria). More precisely, it is a modified version of R commander designed to add statistical functions frequently used in biostatistics [31].

## Results

We identified 15,632 potential participants, of which 1,921 were excluded for not meeting the inclusion criteria or for meeting the exclusion criteria. Thus, a total of 13,711 questionnaires were sent to bereaved family members; 9,123 responses were finally analyzed (effective response rate was 67%). The data analysis included 1,017, 814, and 7,292 family members' questionnaires from home hospice services, acute hospitals, and PCUs, respectively. The mean length of time between patient death and completion of the questionnaire was 283.4 days (standard deviation 139.5 days).

### Characteristics of patients and bereaved family members

Of the 9,123 patients and family caregivers included in this study, 3,841 (42.1%) patients preferred their home for their place of death, 520 (5.7%) preferred an acute-care hospital, 2,754 (30.2%) preferred a PCU, 50 (0.5%) preferred other places (e.g., nursing home), 424 (4.6%) had no preference, 1,187 had unknown preferences (13.0%), and 347 had missing data. The 7,165 patients with a preferred place of death were categorized as follows: 3,640 patients into the achieved group and 3,525 patients into the not-achieved group.

The characteristics of the patients with a preferred place of death and their family caregivers are shown in Table 1.

**Table 1. Characteristics of Patients with a Preferred Place of Death and Their Family Caregivers.**

| Patient and Family Caregiver Characteristics | Total n=7165 | | Achieved n=3640 | | Not achieved n=3525 | | |
|---|---|---|---|---|---|---|---|
| | n | (%) | n | (%) | n | (%) | p value* |
| Patient Characteristics | | | | | | | |
| Gender | | | | | | | 0.58 |
| Male | 3997 | 55.6 | 2018 | 40.0 | 1979 | 38.8 | |
| Female | 3088 | 43.1 | 1580 | 51.2 | 1508 | 48.8 | |
| Age Average: 73.5   SD: 11.5 | | | | | | | p<0.001* |
| <40y | 52 | 0.7 | 28 | 40.6 | 24 | 34.8 | |
| 40-49y | 178 | 2.5 | 84 | 47.2 | 94 | 52.8 | |
| 50-59y | 551 | 7.7 | 276 | 50.1 | 275 | 49.9 | |
| 60-69y | 1646 | 23.0 | 910 | 55.3 | 736 | 44.7 | |
| 70-79y | 2249 | 31.4 | 1188 | 52.8 | 1061 | 47.2 | |
| 80-89y | 2094 | 29.2 | 981 | 46.8 | 1113 | 53.2 | |
| ≥90y | 382 | 5.3 | 163 | 42.7 | 219 | 57.3 | |
| Primary tumor site | | | | | | | 0.35 |
| Lung | 1641 | 22.9 | 815 | 49.7 | 826 | 50.3 | |
| Digestive system | 3333 | 46.5 | 1722 | 51.7 | 1611 | 48.3 | |
| Other | 2180 | 30.4 | 1096 | 50.3 | 1084 | 49.7 | |
| Actual place of death | | | | | | | p<0.001* |
| Home | 933 | 13.2 | 864 | 92.6 | 69 | 7.4 | |
| Acute hospitals | 580 | 8.1 | 160 | 27.6 | 420 | 72.4 | |
| PCU | 5652 | 78.9 | 2616 | 46.3 | 3036 | 53.7 | |
| Patients' preferred place of death | | | | | | | p<0.001* |
| Home | 3841 | 53.6 | 864 | 22.5 | 2977 | 77.5 | |
| Acute hospitals | 520 | 7.3 | 160 | 30.8 | 360 | 69.2 | |
| PCU | 2754 | 38.4 | 2616 | 95.0 | 138 | 5.0 | |
| Others (e.g., nursing home) | 50 | 0.7 | 0 | 0 | 50 | 100.0 | |
| Annual Income during care (JPY) | | | | | | | 0.055 |
| <2,000,000 | 2093 | 29.2 | 1015 | 48.5 | 1078 | 51.5 | |
| 2,000,000-3,999,999 | 2676 | 37.4 | 1385 | 51.6 | 1291 | 48.2 | |
| ≥4,000,000 | 1982 | 27.7 | 1021 | 51.5 | 961 | 48.5 | |
| Medical bills | | | | | | | 0.015 |
| <100,000 | 1937 | 27.3 | 1034 | 53.4 | 903 | 46.6 | |
| 100,000-199,999 | 2284 | 31.9 | 1117 | 48.9 | 1167 | 51.1 | |
| ≥200,000 | 2648 | 37.0 | 1343 | 50.7 | 1305 | 49.3 | |
| Intervention of hospital-based Palliative care team | | | | | | | 0.005* |
| Yes | 5482 | 76.5 | 2838 | 51.8 | 2644 | 48.2 | |
| No | 1046 | 14.6 | 489 | 46.7 | 557 | 53.3 | |
| unkown | 465 | 6.5 | 223 | 48.0 | 242 | 52.0 | |
| Implementation of EOL discussion | | | | | | | 0.28 |
| Yes | 5968 | 83.3 | 3013 | 50.5 | 2955 | 49.5 | |
| No | 1012 | 14.1 | 530 | 52.4 | 482 | 47.6 | |
| | | | | | | | |
| Family Caregiver Characteristics | | | | | | | |
| Gender | | | | | | | 0.003* |
| Male | 2373 | 33.1 | 1137 | 47.9 | 1236 | 52.1 | |

*(Continued)*

Table 1. (Continued)

| Patient and Family Caregiver Characteristics | Total n = 7165 | | Achieved n = 3640 | | Not achieved n = 3525 | | |
|---|---|---|---|---|---|---|---|
| | n | (%) | n | (%) | n | (%) | p value* |
| Female | 4684 | 65.4 | 2446 | 52.2 | 2238 | 47.8 | |
| Age Average: 73.5   SD: 11.5 | | | | | | | p < 0.001* |
| <40y | 267 | 3.9 | 117 | 43.8 | 159 | 59.6 | |
| 40-49y | 854 | 11.9 | 366 | 42.9 | 488 | 57.1 | |
| 50-59y | 1695 | 23.7 | 780 | 46.0 | 915 | 54.0 | |
| 60-69y | 2167 | 30.2 | 1132 | 52.2 | 1035 | 47.8 | |
| 70-79y | 1583 | 22.1 | 899 | 56.7 | 684 | 43.2 | |
| 80-89y | 480 | 6.7 | 283 | 59.0 | 197 | 41.0 | |
| ≥90y | 15 | 0.2 | 10 | 66.7 | 5 | 33.3 | |
| Relationship | | | | | | | p < 0.001* |
| Spouse | 3410 | 47.6 | 1861 | 54.6 | 1549 | 45.4 | |
| Child | 2499 | 34.9 | 1098 | 43.9 | 1401 | 56.1 | |
| Son-/daughter-in-low, Parent, Sibling, other | 1180 | 16.5 | 635 | 53.8 | 545 | 46.2 | |
| Education | | | | | | | 0.73 |
| Elementary school to high school | 4013 | 56.0 | 2030 | 50.6 | 1983 | 49.4 | |
| Vocational school, Junior college, Undergraduate, Graduate | 3037 | 42.4 | 1550 | 51.0 | 1487 | 49.0 | |
| Presence of other caregivers | | | | | | | 0.090 |
| Present | 5176 | 72.2 | 2588 | 50.0 | 2588 | 50.0 | |
| Absent | 1871 | 26.1 | 979 | 52.3 | 892 | 47.7 | |
| Communication about the disease and life with the patient | | | | | | | p < 0.001* |
| Frequently | 3017 | 42.1 | 1725 | 57.2 | 1292 | 42.8 | |
| As needed | 3557 | 49.7 | 1660 | 46.7 | 1897 | 53.3 | |
| Rarely | 537 | 7.5 | 225 | 41.9 | 312 | 58.1 | |
| Family caregivers' preferred place of death | | | | | | | p < 0.001* |
| Home | 2155 | 30.0 | 919 | 42.4 | 1236 | 57.4 | |
| Acute Hospitals | 510 | 7.1 | 163 | 32.0 | 347 | 68.0 | |
| PCU | 4098 | 57.2 | 2414 | 66.6 | 1684 | 48.1 | |
| Others (e.g., nursing home) | 30 | 0.4 | 6 | 20.0 | 24 | 80.0 | |
| Congruence between family caregivers' preferred and actual place of death | | | | | | | p < 0.001* |
| congruence | 4738 | 66.1 | 3150 | 66.5 | 1588 | 33.5 | |
| Not-congruence | 2198 | 30.7 | 410 | 18.7 | 1788 | 81.3 | |

*Differences between the groups were tested using a χ2 test.

EOL: end of life; JPY: Japanese yen; PCU: palliative care unit; SD: standard deviation; y: years.

Totals of some items do not add to 100% owing to missing data.

## Preferred place of death

Of the 7,165 patients with preferred place of death data, the congruence rates between patients' preferred and actual place of death was 50.8%. For patients whose preferred place of death was their home, an acute-care hospital, and a PCU, the congruence rates were 22.5%, 30.8%, and 94.5%, respectively.

The GDI scores according to the congruence between patients' preferred and actual place of death are shown in Table 2.

Of the 7,165 family caregivers of patients with preferred place of death, 2,155 (30.0%) preferred home as the place of death for their patients, 510 (7.1%) preferred an acute-care hospital, 4098 (57.2%) preferred a PCU, and 30 (0.4%)

**Table 2. GDI scores according to congruence between patients' preferred and actual place of death.**

| GDI | Total | | | | | Total | home | | | | | home | hospital | | | | | hospital | PCU | | | | | PCU |
|---|---|---|---|---|---|---|---|---|---|---|---|---|---|---|---|---|---|---|---|---|---|---|---|---|
| | Achieved n=3640 | | Not achieved n=3525 | | | | Achieved n=864 | | Not achieved n=69 | | | | Achieved n=160 | | Not achieved n=420 | | | | Achieved n=2616 | | Not achieved n=3036 | | |
| | Mean | SD | Mean | SD | P value | | Mean | SD | Mean | SD | P value | | Mean | SD | Mean | SD | P value | | Mean | SD | Mean | SD | P value |
| GDI | 48.8 | 10.1 | 44.0 | 9.5 | p<0.001* | | 49.0 | 11.8 | 48.2 | 9.6 | 0.55 | | 43.6 | 12.3 | 41.2 | 11.5 | 0.034 | | 49.1 | 9.2 | 44.2 | 9.1 | p<0.001* |
| Physical and psychological comfort | 5.3 | 1.4 | 4.9 | 1.5 | p<0.001* | | 5.1 | 1.5 | 5.0 | 1.6 | 0.76 | | 4.5 | 1.6 | 4.3 | 1.6 | 0.13 | | 5.3 | 1.3 | 5.1 | 1.4 | p<0.001* |
| Dying in a favorite place | 5.7 | 1.3 | 4.1 | 1.6 | p<0.001* | | 6.2 | 1.2 | 5.7 | 1.3 | p<0.001* | | 4.9 | 1.3 | 4.0 | 1.8 | p<0.001* | | 5.6 | 1.2 | 4.2 | 1.6 | p<0.001* |
| Maintaining hope and pleasure | 4.6 | 1.6 | 3.9 | 1.6 | p<0.001* | | 4.9 | 1.5 | 4.8 | 1.7 | 0.42 | | 3.9 | 1.6 | 3.6 | 1.7 | 0.088 | | 4.6 | 1.6 | 3.9 | 1.6 | p<0.001* |
| Good relationship with medical staff | 5.8 | 1.1 | 5.3 | 1.3 | p<0.001* | | 5.8 | 1.2 | 5.9 | 1.0 | 0.87 | | 5.6 | 1.2 | 5.3 | 1.3 | 0.014 | | 5.8 | 1.1 | 5.3 | 1.3 | p<0.001* |
| Not being a burden to others | 3.5 | 1.6 | 3.5 | 1.6 | 0.86 | | 3.3 | 1.6 | 3.5 | 1.6 | 0.17 | | 3.6 | 1.6 | 3.4 | 1.5 | 0.37 | | 3.5 | 1.6 | 3.5 | 1.6 | 0.27 |
| Good relationship with family | 5.3 | 1.4 | 4.8 | 1.5 | p<0.001* | | 5.5 | 1.3 | 5.0 | 1.5 | 0.002 | | 4.9 | 1.5 | 4.5 | 1.5 | 0.010 | | 5.2 | 1.4 | 4.8 | 1.4 | p<0.001* |
| Independence | 3.4 | 1.9 | 2.9 | 1.9 | p<0.001* | | 3.4 | 1.9 | 3.1 | 1.8 | 0.20 | | 3.6 | 1.9 | 3.5 | 1.9 | 0.71 | | 3.4 | 1.9 | 2.8 | 1.8 | p<0.001* |
| Environmental comfort | 5.6 | 1.1 | 5.1 | 1.3 | p<0.001* | | 5.7 | 1.2 | 5.4 | 1.1 | 0.076 | | 5.0 | 1.3 | 4.6 | 1.5 | 0.002 | | 5.6 | 1.1 | 5.2 | 1.3 | p<0.001* |
| Being respected as an individual | 6.1 | 0.9 | 5.9 | 1.1 | p<0.001* | | 6.2 | 0.9 | 6.0 | 1.0 | 0.092 | | 5.7 | 1.2 | 5.6 | 1.2 | 0.33 | | 6.1 | 0.9 | 5.9 | 1.1 | p<0.001* |
| Life completion | 4.9 | 1.8 | 4.4 | 1.8 | p<0.001* | | 4.9 | 1.8 | 5.0 | 1.6 | 0.710 | | 4.4 | 1.8 | 4.1 | 1.8 | 0.12 | | 4.9 | 1.7 | 4.5 | 1.8 | p<0.001* |

GDI: Good Death Inventory; PCU: palliative care unit; SD: standard deviation.

preferred others (e.g., nursing home), 133 (1.9%) had no preference, 199 had unknown preferences (2.8%), and 40 had missing data. The congruence rates between family caregivers' preferred place of death and actual place of death was 66.1%. For caregivers whose preferred place of death for their patients was home, an acute-care hospital, and a PCU, the congruence rates were 34.1%, 36.6%, and 93.1%, respectively (data not shown).

## Congruence between patients' preferred and actual place of death and its association with quality of death and dying

The mean GDI scores were 50.8±7.9, 43.5±9.1, and 46.9±8.4 for home, acute-care hospital, and PCU deaths, respectively.

Fig 1 and Table 2 show the mean GDI scores for both the achieved group and the not-achieved group. With respect to the congruence between preferred and actual place of death, GDI scores were significantly higher for the achieved group than the not-achieved group (48.8±10.1 and 44.0±9.5, respectively; p<0.001).

For those who died in acute-care hospitals or PCUs, GDI scores were significantly higher among the achieved group than the not-achieved group. For those who died at home, nonsignificant differences were observed in GDI scores between groups.

A significant difference between groups was observed (achieved, not-achieved, no preference, and preference unknown), as determined by a one-way ANOVA (p<0.001). A post hoc Dunnett test showed that GDI scores were significantly higher for achieved compared with not-achieved, no preference, and preference unknown for preferred place of death (see S1 Table).

## Domains of GDI according to congruence between patients' preferred and actual place of death

All domains, aside from "not being a burden to others," scored higher in the achieved group compared with the not-achieved group. Domain scores according to care setting are summarized in Fig 2 and S1–S3 Figs.

## Factors related to quality of death and dying: Results of a multiple linear regression analysis

Multiple linear regression analysis was performed with the aforementioned analyzed variables included as independent variables. The GDI score was used as the dependent variable. Table 3 shows the results of the multiple linear regression analysis. Congruence between the patient's preferred and actual place of death, intervention of a hospital-based palliative care team, the presence of other caregivers, frequent communication about the disease and life with the patient, and congruence between the family caregiver's preferred and actual place of death were all associated with higher GDI scores.

## Discussion

This nationwide survey, which was conducted in Japan, showed how congruence between a cancer patient's preferred and actual place of death affected their quality of death and dying.

Our findings indicate that cancer patients who died in the place they preferred had better quality of death and dying than patients who died in a place they did not prefer, as reported by their family caregivers.

This study used the short version of the GDI to measure quality of death and dying of terminally ill patients, and confirmed that congruence between preferred and actual place of death was strongly related to family-reported quality of death and dying. We found that quality of death and dying was higher in the achieved group than the not-achieved group. Moreover, the findings show that a better quality of death and dying was achieved in many components of the GDI (i.e., physical and psychological comfort, maintaining hope and pleasure, good relationship with medical staff, spending enough time with family, environmental comfort, being respected as an individual, and life completion).

Previous studies have shown that patients' quality of death and dying was significantly higher at home relative to other places [12–14]. Our results were similar.

*Differences between the groups were tested using t-tests. PCU: Palliative Care Unit.

**Fig 1. GDI Scores According to Congruence Between Patients' Preferred and Actual Place of Death.** *Differences between the groups were tested using t-tests. PCU: palliative care unit.

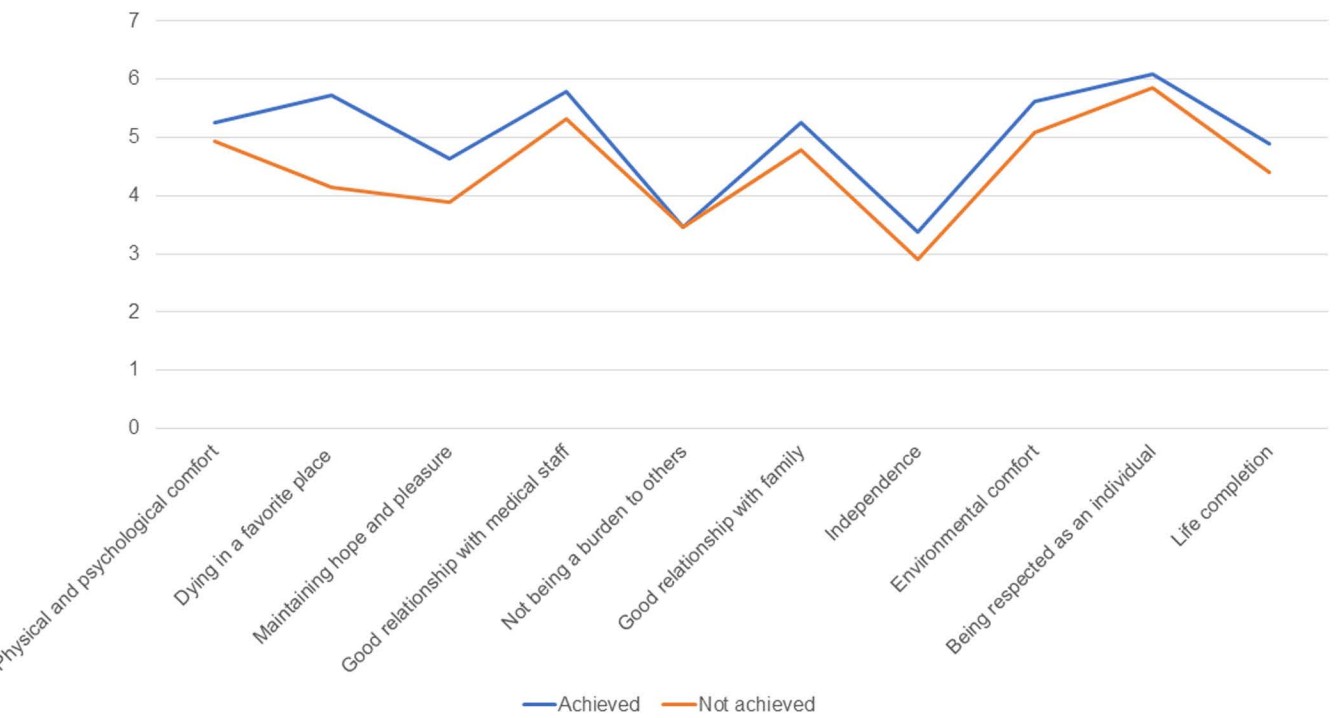

**Fig 2. GDI Domain Scores According to Congruence Between Patients' Preferred and Actual Place of Death.**

**Table 3. Factors related to quality of death: results of a multiple linear regression analysis.**

| | β | 95%CI | P value |
|---|---|---|---|
| Patient characteristics | | | |
| Gender | | | |
| Male (reference) | 0 | | |
| Female | −0.27 | −0.90 to 0.26 | 0.32 |
| Age | 0.08 | 0.06 to 0.11 | p<0.001* |
| Primary tumor site | | | |
| Lung (reference) | 0 | | |
| Digestive system | 0.38 | −0.20 to 0.96 | 0.20 |
| Other | 0.33 | −0.31 to 0.97 | 0.31 |
| Actual place of death | | | |
| Home (reference) | 0 | | |
| Acute hospitals | −3.82 | −5.36 to −2.70 | p<0.001* |
| PCU | −0.89 | −1.61 to −0.16 | 0.016 |
| Congruence between preferred and actual place of death | | | |
| Achieved (reference) | 0 | | |
| Not achieved | −3.60 | −4.72 to −3.06 | p<0.001* |
| Annual Income during care (JPY) | | | |
| <2,000,000 (reference) | 0 | | |
| 2,000,000-3,999,999 | 0.19 | −0.30 to 0.75 | 0.64 |
| ≥4,000,000 | 0.68 | 0.16 to 1.30 | 0.029 |
| Medical bills | | | |
| <100,000 (reference) | 0 | | |
| 100,000-199,999 | −0.44 | −0.47 to 0.15 | 0.14 |
| ≥200,000 | −0.89 | −1.59 to −0.30 | p<0.001* |
| Intervention of Hospital-based Palliative care team | | | |
| Yes (reference) | 0 | | |
| No | −1.07 | −1.69 to −0.42 | 0.001* |
| unkown | −2.19 | −3.16 to −1.21 | p<0.001* |
| Implementation of EOL discussion | | | |
| Yes (reference) | 0 | | |
| No | −0.70 | −1.37 to −0.01 | 0.045 |
| Gender | | | |
| Male (reference) | 0 | | |
| Female | −0.48 | −1.03 to 0.07 | 0.089 |
| Age | 0.02 | −0.01 to 0.05 | 0.13 |
| Relationship | | | |
| Spouse (reference) | 0 | | |
| Child | −0.008 | −0.86 to 0.84 | 0.99 |
| Son-/daughter in low, Parent, Sibling, other | 0.76 | 0.03 to 1.50 | 0.042 |
| Education | | | |
| Elementary school, high school (reference) | 0 | | |
| Vocational school, junior college, Undergraduate, Graduate | −0.23 | −0.72 to 0.26 | 0.35 |
| Presence of other caregivers | | | |
| Present (reference) | 0 | | |

*(Continued)*

**Table 3.** (Continued)

|  | β | 95%CI | P value |
|---|---|---|---|
| Absent | −1.39 | −1.86 to −0.87 | p<0.001* |
| Communication about the disease and life with the patient |  |  |  |
| Frequently (reference) | 0 |  |  |
| As needed | −2.04 | −2.52 to −1.56 | p<0.001* |
| Rarely | −3.14 | −4.05 to −2.22 | p<0.001* |
| Congruence between family caregivers' preferred and actual place of death |  |  |  |
| Congruence | 0 |  |  |
| Not-Congruence | −1.51 | −2.07 to −0.96 | p<0.001* |

β: partial regression coefficient; CI: confidence interval; EOL: end of life; JPY: Japanese yen; PCU: palliative care unit.

Adjusted R-squared, which is an indicator of effect size, was 0.12; the standard for effect size was moderate.

Those who died in acute-care hospitals and PCUs had GDI scores that were significantly higher for the achieved group compared with the not-achieved group. Nonsignificant differences were observed for those who died at home. This may be attributable to the smaller difference between the achieved and not-achieved groups in the GDI domains for the home setting than those for hospitals or PCUs.

In the present study, several factors were identified as affecting the congruence between patients' preferred and actual place of death: actual place of death, patients' preferred place of death, intervention of a hospital-based palliative care team, the presence of other caregivers, communication about the disease and life with the patient, and congruence between family caregivers' preferred and actual place of death. In previous studies, similar results have been obtained. It has been reported that the factors associated with congruence between patients' preferred and actual place of death include patients providing their opinions regarding where they want to die, [32–34] patient–caregiver agreement on the preferred place of death [4], and the presence of a caregiver [26,33,34].

This study had several limitations. First, the participants were recruited from facilities that were members of HPCJ, which has a high percentage of PCU members. The proportion of patients who died in PCUs in the present study was therefore relatively high, and the findings may not wholly represent the extent of palliative care services in Japan. Second, we could not exclude recall bias. According to previous studies, we consider that 3–12 months after bereavement may be an appropriate time frame for the inclusion criteria regarding both recall bias and the grieving process [20–24]. Despite these limitations, the results of this study offer important insights and practical guidance for the period of time in which cancer patients and their families determine the place for end-of life care and place of death. Future research is needed to determine why a preferred place of death results in better end-of-life quality for patients.

## Conclusion

Congruence between preferred and actual place of death was associated with better quality of death and dying among terminally ill patients with cancer. This factor should be regarded as one of the essential components of end-of-life care.

## Supporting information

**S1 Fig. GDI Domain Scores for Death at Home.**
(TIF)

**S2 Fig. GDI Domain Scores for Death in an Acute-Care Hospital.**
(TIF)

**S3 Fig. GDI Domain Scores for Death in a PCU.**
(TIF)

**S1 Table. GDI Scores for Congruence Between Preferred and Actual Place of Death Across Groups.** GDI: Good
Death Inventory.
(XLSX)

## Acknowledgments

We thank Anita Harman, PhD, and Bronwen Gardner, PhD, from Edanz (https://jp.edanz.com/ac) for editing a draft of this
manuscript.

## Author contributions

**Conceptualization:** Mariko Shutoh, Tatsuya Morita.

**Data curation:** Maho Aoyama.

**Formal analysis:** Mariko Shutoh.

**Investigation:** Mariko Shutoh.

**Methodology:** Mariko Shutoh, Tatsuya Morita.

**Supervision:** Tatsuya Morita, Maho Aoyama, Mitsunori Miyashita.

**Validation:** Mariko Shutoh.

**Writing – original draft:** Mariko Shutoh.

**Writing – review & editing:** Tatsuya Morita, Maho Aoyama, Yoshiyuki Kizawa, Yasuo Shima, Mitsunori Miyashita.

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
