## [Decision Letter · Decision Letter 0]

Dear Dr. Shutoh,

Thank you for submitting your manuscript to PLOS ONE. After careful consideration, we feel that it has merit but does not fully meet PLOS ONE’s publication criteria as it currently stands. Therefore, we invite you to submit a revised version of the manuscript that addresses the points raised during the review process.

**ACADEMIC EDITOR: Please could the changes suggested by the reviewers be addressed. The reviewers have requested clarification in sections. The focus on patients with cancer also needs to be stated in the title. Pl**
**ease also address Reviewer 1's suggestion to include references to recent studies as you see fit as they seem relevant to the paper.**

We look forward to receiving your revised manuscript.

Kind regards,

Gursharan K Singh, PhD

Academic Editor

PLOS ONE

For additional information about PLOS ONE ethical requirements for human subjects research, please refer to http://journals.plos.org/plosone/s/submission-guidelines#loc-human-subjects-research .

Reviewers' comments:

Reviewer's Responses to Questions

**Comments to the Author**

1. Is the manuscript technically sound, and do the data support the conclusions?

Reviewer #1: Yes

Reviewer #2: Yes

2. Has the statistical analysis been performed appropriately and rigorously?

Reviewer #1: Yes

Reviewer #2: Yes

3. Have the authors made all data underlying the findings in their manuscript fully available?

Reviewer #1: Yes

Reviewer #2: Yes

4. Is the manuscript presented in an intelligible fashion and written in standard English?

Reviewer #1: Yes

Reviewer #2: No

Reviewer #1: Thank you for the opportunity to review your paper “Congruence between preferred and actual place of death and its association with patients’ quality of death and dying: A nationwide survey in Japan”. This is a very important topic, which creates opportunities for improving care at the end of life.

My main suggestions include:

The manuscript is well-written and easy to follow. Really valuable and interesting results, further supporting the importance of end of life care planning and striving to fulfil patients’ preferences for place of care and death, to achieve qualitative care at the end of life. This study only focuses on patients with cancer, right? If so then this should be made clear in the title and the aim, and a motivation to why the study is limited to only patients with cancer and not other chronic illness with potential palliative care would be good.

Abstract:

- The abstract is informative and contains the essential information from the study.

Introduction:

- The introduction is easy to follow and contains adequate parts, and leads onto a rational and the research questions. The introduction could be developed and present more about common preferences for care at the end of life and place of death as well as quality indicators and qualitative care at the end of life. Pls see some recommended references below.

Methods:

- The methods section is well-written and clear.

- Good sample size, sound data and choice of methods.

- Data collection, how long after the death did the bereaved family members respond to the survey, was there any reasoning about when to send the survey invite? I see this is mentioned only as a limitation with regard to recall bias, a descroiåiption under methods would be helpful too. This is important for several reasons, firstly not too soon for ethical reasons of potentially upsetting the family members, secondly not too close to the yearly anniversary of the death for the same reason, thirdly the question of recall bias, if too long time has passed since the death.

- Setting, a setting description would be helpful describing the type of care available and the end of life in Japan and how healthcare is organised in Japan, if it is universal or not. There is a really high proportion of the patients that had died in PCU compared to results from place of death other countries, see examples of references for Europe below.

- Previous research has shown how the preference may change closer to death and how the last days or week may be spend in the preferred place, but that the actual death may take place in hospital or hospice care. Could you differentiate the place for care the last few days and the place of death or did family members only give one response for last few days of care and death together?

Results:

- Is there any information about the non-responders and if they differ from the responders?

- In total 11% died at home, and the preferred place of at home for death for patients was 40% and 26% for family members’ preferences. That is an interesting differences between actual place of death, and preferences, as well as the difference in preference between patients and family members.

- Intervention of palliative care team lower % than the 80% dying in PCU for achieved place, doesn’t PCU mean an automatic intervention of a palliative care team?

- Table 1, under relationship total there is no % presented for son, daughter in law etc…The % for education total most be wrong, does not add up to 100?

- Row 185: maybe state the place in connection with the % for this sentence “The congruence rates between caregivers’ preferred and actual place of death for their loved ones were 32.1%, 93.2%, and 38.7% for home, PCU, and acute-care hospital, respectively”.

- It would be good if you could clearer describe that the preferred place for care and death is the patient’s preference as reported by the family member as a proxy for the patient, but also that you asked about the family member’s preferences for where the patient should be care for and place of death, if I have understood this correctly as described under the results? This could be made clearer under data collection.

Discussion:

- Good summary of the results, and a well-written discussion which raises the most essential results.

- The differences between patients’ and family members’ preferences. Congruences between family members’ preferences and actual place of death?

Conclusion:

- In the future extend research to include other diagnoses with potential palliative care needs than only patients with cancer?

There are more recent studies on place of death as well as preferences thereof, the introduction and discussion could be strengthened with more reference and description of an international context.

Example studies place of death Europe:

Öhlén, J., Stina, N., Anneli, O., Stefan, N., Hanna, G., Johan, F. C., & Cecilia, L. (2025). Influence of palliative care policy on place of death for people with different cancer types: a nationwide’register study. PloS one, 20(3), e0320086.

Cohen, J., Pivodic, L., Miccinesi, G., Onwuteaka-Philipsen, B. D., Naylor, W. A., Wilson, D. M., ... & Deliens, L. (2015). International study of the place of death of people with cancer: a population-level comparison of 14 countries across 4 continents using death certificate data. British journal of cancer, 113(9), 1397-1404.

O'Sullivan, A., Larsdotter, C., Sawatzky, R., Alvariza, A., Imberg, H., Cohen, J., & Öhlén, J. (2024). Place of care and death preferences among recently bereaved family members: a cross-sectional survey. BMJ Supportive & Palliative Care, 14(e3), e2904-e2913.

Cross, S. H., & Warraich, H. J. (2019). Changes in the place of death in the United States. New England Journal of Medicine, 381(24), 2369-2370.

Reviewer #2: Thank you for the opportunity to review the manuscript titled 'Congruence between preferred and actual place of death and its association with patients’ quality of death and dying: A nationwide survey in Japan.' The topic provides relevant information on factors influencing the quality of death. Please find my comments in the attached file.

**Do you want your identity to be public for this peer review?** For information about this choice, including consent withdrawal, please see our Privacy Policy

Reviewer #1: **Yes: ** Anna O'Sullivan

Reviewer #2: No

---

## [Author Response · Author response to Decision Letter 1]

4 Jun 2025

Gursharan K Singh, PhD

Academic Editor

PLOS ONE

04 June 2025

Dear Dr Singh,

Re: Manuscript reference No. PONE-D-25-05638

Please find attached a revised version of our manuscript “Congruence between preferred and actual place of death and its association with advanced cancer patients’ quality of death and dying: A nationwide survey in Japan,” which we would like to resubmit for publication as an Original Article in PLOS ONE.

Your comments and those of the reviewers were highly insightful and enabled us to greatly improve the quality of our manuscript. In the following pages, please find our point-by-point responses to each of the reviewers’ comments as well as your own comments.

Revisions in the text are shown using yellow highlight for additions and strikethrough font for deletions. In accordance with a suggestion from Reviewer 1, we have stated the focus on patients with cancer in the title and included references to recent studies. We hope that the revisions in the manuscript and our accompanying responses will be sufficient to make our manuscript suitable for publication in PLOS ONE.

We look forward to hearing from you at your earliest convenience.

Yours sincerely,

Dr Mariko Shutoh

Minato Home Care Clinic, Chuo-ku, Tokyo, Japan

Ph: +81 358590812

Fax: +81 358590822

E-mail: shut44105m@gmail.com

---

## [Decision Letter · Decision Letter 1]

Congruence between preferred and actual place of death and its association with quality of death and dying in advanced cancer patients: A nationwide survey in Japan

PONE-D-25-05638R1

Dear Dr. Shutoh,

We’re pleased to inform you that your manuscript has been judged scientifically suitable for publication and will be formally accepted for publication once it meets all outstanding technical requirements.

Kind regards,

Gursharan K Singh, PhD

Academic Editor

PLOS ONE

Additional Editor Comments (optional):

Reviewers' comments:

Reviewer's Responses to Questions

**Comments to the Author**

Reviewer #1: All comments have been addressed

Reviewer #2: All comments have been addressed

2. Is the manuscript technically sound, and do the data support the conclusions?

Reviewer #1: Yes

Reviewer #2: Yes

3. Has the statistical analysis been performed appropriately and rigorously?

Reviewer #1: Yes

Reviewer #2: Yes

4. Have the authors made all data underlying the findings in their manuscript fully available?

Reviewer #1: Yes

Reviewer #2: Yes

5. Is the manuscript presented in an intelligible fashion and written in standard English?

Reviewer #1: Yes

Reviewer #2: Yes

Reviewer #1: Thank you for considering my suggestions and making changes to your manuscript accordingly. I find all revisions that you have made satisfactory. Your manuscript is now significantly clearer, and I recommend it for publication.

Reviewer #2: Thank you for addressing my previous comments. I truly appreciate your hard work and thoughtful contribution to this important topic.Thank you again for your valuable work and contribution to the field.

**Do you want your identity to be public for this peer review?** For information about this choice, including consent withdrawal, please see our Privacy Policy

Reviewer #1: **Yes: ** Anna O'Sullivan

Reviewer #2: No

---

## [Editor Report · Acceptance letter]

PONE-D-25-05638R1

PLOS ONE

Dear Dr. Shutoh,,

I'm pleased to inform you that your manuscript has been deemed suitable for publication in PLOS ONE. Congratulations! Your manuscript is now being handed over to our production team.

Kind regards,

on behalf of

Dr. Gursharan K Singh

Academic Editor

PLOS ONE